# Neural Taskonomy: Inferring the Similarity of Task-Derived Representations from Brain Activity

**Aria Y. Wang**
Carnegie Mellon University
`ariawang@cmu.edu`

**Michael J. Tarr**
Carnegie Mellon University
`michaeltarr@cmu.edu`

**Leila Wehbe**
Carnegie Mellon University
`lwehbe@cmu.edu`

## Abstract

Convolutional neural networks (CNNs) trained for object classification have been widely used to account for visually-driven neural responses in both human and primate brains. However, because of the generality and complexity of object classification, despite the effectiveness of CNNs in predicting brain activity, it is difficult to draw specific inferences about neural information processing using CNN-derived representations. To address this problem, we used learned representations drawn from 21 computer vision tasks to construct encoding models for predicting brain responses from BOLD5000—a large-scale dataset comprised of fMRI scans collected while observers viewed over 5000 naturalistic scene and object images. Encoding models based on task features predict activity in different regions across the whole brain. Features from 3D tasks such as keypoint/edge detection explain greater variance compared to 2D tasks—a pattern observed across the whole brain. Using results across all 21 task representations, we constructed a "task graph" based on the spatial layout of well-predicted brain areas from each task. A comparison of this brain-derived task structure to the task structure derived from transfer learning accuracy demonstrate that tasks with higher transferability make similar predictions for brain responses from different regions. These results—arising out of state-of-the-art computer vision methods—help reveal the task-specific architecture of the human visual system.

## 1   Introduction

Scene understanding requires the integration of space perception, visual object recognition, and the extraction of semantic meaning. The human brain's solution to this challenge has been elucidated in recent years by the identification of scene-selective brain areas via comparisons between images of places and common objects[1]. This basic contrast has been extended across a wide variety of image manipulations that have provided evidence for the neural coding of scene-relevant properties such as relative openness[2–4], the distance of scenes to the viewer[2,3,5], 3D spatial layout[6–8] and navigational affordances[9]. Recently, to help explain such findings, Lescroart and Gallant[10] developed an encoding model using a feature space that parametrizes 3D scene structures along the distance and orientation dimensions and provides a computational framework to account for human scene processing. Intriguingly, Lescroart and Gallant[10] were able to identify distance and openness within scenes as the dimensions that best account for neural responses in scene-selective brain areas. At a higher, semantic, level, Stansbury et al.[11] found that neural responses in scene-selective brain areas can be predicted using scene categories that were learned from object co-occurrence statistics. Such findings demonstrate that human scene-selective areas represent both visual and semantic scene features. At the same time, there is still no robust model of how these different kinds of information are integrated both within and across brain regions.

Encoding models are widely used in understanding feedforward information processing in human perception, including scene perception. Encoding models are predictive models of brain activity that are able to generalize and predict brain responses to novel stimuli[12]. Researchers have also used encoding models to infer which dimensions are critical for prediction by comparing the weights learned by the model[10,13]. One of the successes of encoding models lies in predicting low- to mid-level visual cortex responses in humans and primates using features that were learned via a convolutional neural network trained on object recognition[14–17]. Most interestingly, these studies demonstrate a correspondence between human neural representation and learned representations within CNN models along the perceptual hierarchy: early layers tend to predict early visual processing regions, whereas later layers tend to predict later visual processing regions. Similarly, researchers have found that network representations from other task-driven networks, including networks trained on speech or music related tasks, are able to explain neural responses in human auditory pathways[18]. Such successes are not mere coincidences but rather indications of how fundamental task-driven representations are to both task training and to information processing in the brain.

Despite these advances, CNN features themselves are notoriously difficult to interpret. First, activations from the convolutional layers lie in extremely high-dimensional spaces and it is difficult to interpret what each feature dimension signifies. Second, features from a CNN tailored for a particular visual task can represent any image information that is relevant to that task. As a consequence of these two issues, the feature representations learned by the network are not necessarily informative with respect to the nature of visual processing in the brain despite their good performance in predicting brain activity.

To better understand the specificity of the information represented in the human visual processing pathways, we adopted a different approach. Instead of choosing a generic object-classification CNN as a source of visual features, we built encoding models with individual feature spaces obtained from different task-specific networks. These tasks included mid-level features such as surface normal estimation, edge detection, scene classification, etc. In any task-driven network, the feature space learned to accomplish the task at hand should only represent information from input images that is task-relevant. Therefore we can use the predictive regions from each of the models to identify the brain regions where specific task-relevant information is localized. Independently, Dwivedi and Roig[19] have shown that representation similarity analysis (RSA) performed between task representations and brain representations can differentiate scene-selective regions of interest (ROIs) by their preferred task. For example, representations in scene-selective occipital place area (OPA) are more highly correlated with representations from a network trained to predict navigational affordances. However, this study was limited to pre-defined regions of interest, while the task representations we identify span the entire brain. Consequently, the brain regions predicted by each model provide an atlas of neural representation of visual tasks and allow us to further study the representational relationships among tasks.

Independently of the brain, visual tasks have relationships among them. Task representations that are learned specifically for one task can be transferred to other tasks. Computer vision researchers commonly use transfer learning between tasks to save supervision and computational resources. In this vein, Zamir et al.[20] recently showed that by standardizing model structure and measuring performance in transfer learning, one can generate a taxonomic map for task transfer learning ("Taskonomy"). This map provides an account of how much information is shared across different vision tasks. Given this global task structure, we can infer clusters of information defined by segregation of tasks, and then ask: does the brain represent visual information in the same task-relevant manner?

We compared the relationships between tasks using both brain representations and task learning. These comparisons reveal clustering of 2D tasks, 3D tasks, and semantic tasks. Compared to general encoding models, building individual encoding models and exploiting existing relationship among models has the potential to provide more in-depth understanding of the neural representation of visual information.

## 2  Methods[1]

### 2.1  Encoding Model

To explore how and where visual features are represented in human scene processing, we extracted different features spaces describing each of the stimulus images and used them in an encoding model to predict brain responses. Our reasoning is as follows. If a feature is a good predictor of a specific brain region, information about that feature is likely encoded in that region. In this study, we first parameterized each image in the training set into values along different feature dimensions in a feature space. For example, if the feature space of interest is an intermediate layer in a task-driven network, we simply fed the image into the network and extracted its layer activation. These values are used as regressors in a ridge regression model (implemented in PyTorch; see[21]) to predict brain responses to that image. Performance from the validation data is used to choose the regularization parameter in the ridge regression model. We chose to use a ridge regression model instead of more complicated models in order to retain the interpretability of model weights, which may provide insights into the underlying dimensions of the brain responses. For each subject, each voxel's regularization parameter was chosen independently via 7-fold cross-validation based on the prediction performance of the validation data. Model performance was evaluated on the test data using both Pearson's correlation and coefficient of determination ($R^2$). To determine the significance of the predictions, we ran permutation tests where we shuffled responses 5000 times, computed the correlation scores, and obtained FDR corrected $p$-values for both ROI and whole brain results.

### 2.2  Feature Spaces

To simultaneously test representations from multiple 2D, and 3D vision tasks, we used the latent space features from each of the 21 tasks in Taskonomy[20] model bank: autoencoding, colorization, curvature estimation, denoising, depth estimation, edge detection (2D), edge detection (3D) or occlusion edges detection, keypoint detection (2D), keypoint detection (3D), depth, reshading, room layout estimation, segmentation (2D), segmentation (2.5D), surface normal estimation, vanishing point estimation, semantic segmentation, jigsaw puzzle, inpainting, object classification and scene classification. In the Taskonomy training scheme, an intermediate latent space with fixed dimension ($16 \times 16 \times 8$) was enforced for each of these networks. We obtained these latent space activations by feeding our images into each pre-trained task-specific network in the task bank provided with the Taskonomy paper. Four of the 25 tasks were excluded from this analysis because these tasks take multiple images as input, while the brain responses we have are only to single images. Examples of these excluded tasks include camera pose estimation and egomotion estimation. We then built individual ridge regression models with the extracted latent features to predict brain responses and measured the correlation between the prediction and the true response in the held-out dataset.

### 2.3  Neural Data

The images used in this paper are from a publicly available large-scale fMRI dataset, BOLD5000[22]. In the BOLD5000 study, participants' brains were scanned while they fixated at real-world images and judged how much they liked the image using a button press. Images in the BOLD5000 dataset were chosen from standard computer vision datasets (ImageNet[23], COCO[24] and SUN[25]). The experiment was run in a slow-event setting where trials are separated by 10 seconds. From BOLD5000, we used data from three participants viewing 4916 unique images. These 4916 image trials are separated into random training, validation, and testing sets during model fitting. Average of TR 3 and 4 of each slow-event trial is used for model fitting and testing. Region of interest (ROI) boundaries that identify category-selective brain regions in the whole-brain map presented in our results were generated directly from the ROI masks provided with the BOLD5000 dataset.

### 2.4  Task Similarity Computation

For each task, we took prediction performance scores across all voxels ($n \approx 55,000$). We set the score of a voxel to zero if the $p$-value of the correlation obtained from permutation test is above significance threshold ($p > 0.05$, FDR corrected). This gave us a performance matrix of meaningful

correlations of size $m \times v$, where $m$ is the number of tasks of interest and $v$ is the number of voxels. To analyze the relationship between tasks based on neural representations, we computed pairwise similarity across tasks in the performance matrix using cosine similarity. These pairwise similarities were then used to construct graphs and similarity trees among tasks. Other distance or similarity functions such as euclidean distance did not show substantial differences.

## 3 Results

### 3.1 Model Prediction on ROIs

In Figure 1 we show the prediction accuracy measured using the Pearson correlation coefficient. This was done for the 21 task-related feature spaces that were used to predict brain responses in predefined ROIs. Each bar shown in the figure represents the average correlation score across all voxels in that ROI. Overall, the predictions using these feature spaces—which come from mid-level computer vision tasks—show significant correlations with brain responses, except for the feature space from the curvature task. Among scene-selective regions, such as parahippocampal place area (PPA), retrosplenial complex (RSC), occipital place area (OPA), and lateral occipital complex (LOC), models with 3D features (e.g. keypoints, edges) show far better predictions than models with 2D features. This finding is consistent with the results of Lescroart and Gallant[10]. In contrast, within early visual areas, the prediction results between 2D and 3D features are not differentiable. Across all ROIs, features from object and scene classification tasks provide the best predictions. For more scene specific tasks or semantic tasks such as 3D keypoints/edges, 2.5D/semantic segmentation, depth, distance, reshading, surface normal, room layout, vanishing points estimation, and object/scene classification, scene-selective regions are better predicted as compared to early visual areas. These patterns are consistent across all three participants. To quantify the consistency of results across subjects, we computed correlations of prediction accuracy for each pair of subjects: 0.7957 (S1 vs. S2), 0.9034 (S1 vs. S3) and 0.9345 (S2 vs. S3). These results provide evidence that scene-selective areas show selectivity for scene-specific task representations.

### 3.2 Model Prediction Across the Whole Brain

Prediction performance in pre-defined ROIs may omit relevant information arising in other brain regions. In Figure 2 and 3 we show prediction performance across the entire brain in a flattened view (generated using Pycortex[26]). Figure 2 shows the raw prediction performance as correlation coefficients for each task feature space. Figure 3 shows a contrast in prediction between 3D and 2D keypoints as well as edges. In this figure, red-colored voxels are better predicted by 3D features than 2D features, and vice-versa for blue-colored voxels; white-colored voxels are well predicted by both features. We find that 3D features make better predictions for scene-selective regions—those delimited by ROI borders, while 3D and 2D features seem to predict early visual areas equally well. Figure 2 shows that prediction results are consistent across three participants despite anatomical differences in their brain structures and 3 shows that the results are consistent across tasks.

Model performance using feature spaces from other tasks are shown in Figure 4. Here we plot 6 of the 21 tasks, and the remaining figures for this sample subject (subject 1) are provided in the appendix. Voxels with insignificant predictions ($p \geq 0.05$, FDR corrected) are masked in these figures. Prediction performances of all tasks and all three subjects can be viewed at https://cs.cmu.edu/~neural-taskonomy.

To provide a better estimate of the variance ceiling, we ran ridge regression to predict responses of one subject from another. In Figure 5 we show the prediction correlation for each subject from the remaining two subjects. The average correlation between predictions and true responses across voxels for each subject are: 0.0931, 0.0932, 0.112, as shown by the black lines on each plot. The histogram includes low signal to noise ratio (SNR) voxels that are not engaged by the task. The histograms distribution indicates that the accuracy we obtained on significant voxels across the whole brain using features from various task is close to the ceiling. Cross subject prediction results from each pair of subjects is provided in the appendix. Note that we are predicting single-trial fMRI data with no repetitions, which leads to a lower signal to noise ratio (and therefore lower variance ceiling) than other fMRI studies that average repetitions.

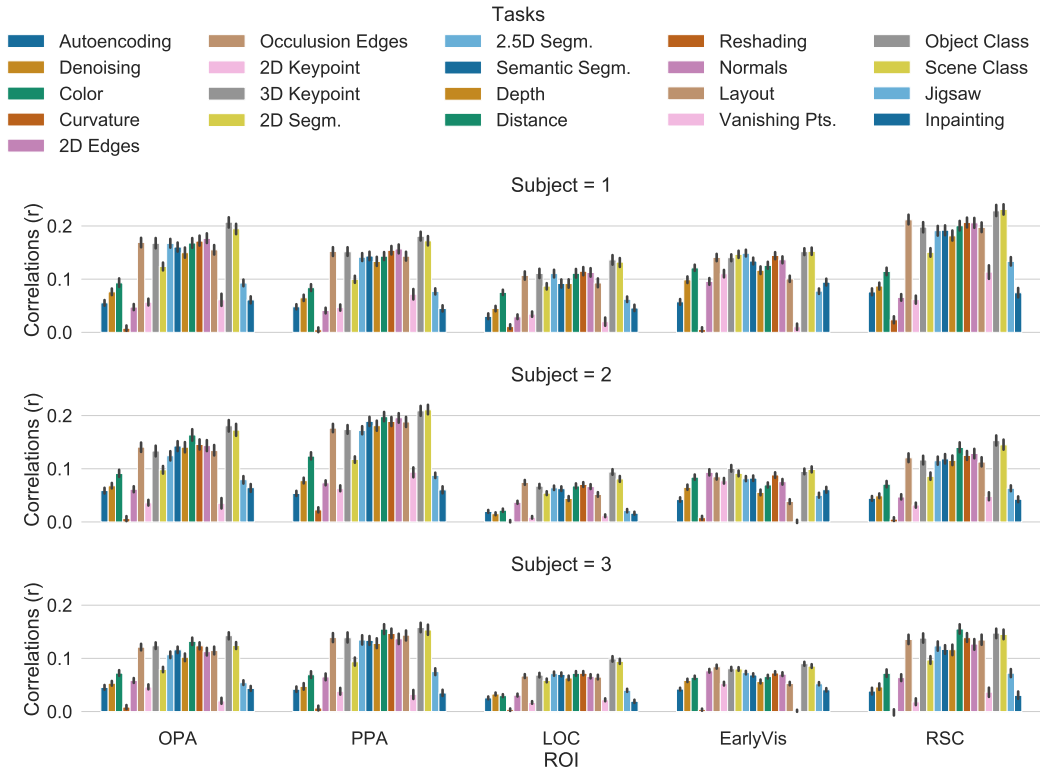

Figure 1: **Pearson correlation coefficient between predicted and true responses across tasks.** Each sub-figure corresponds to a particular participant. Colors in the legend are arranged by columns. Features from 3D tasks, compared to those from 2D tasks, predict better in OPA, PPA, RSC, and LOC.

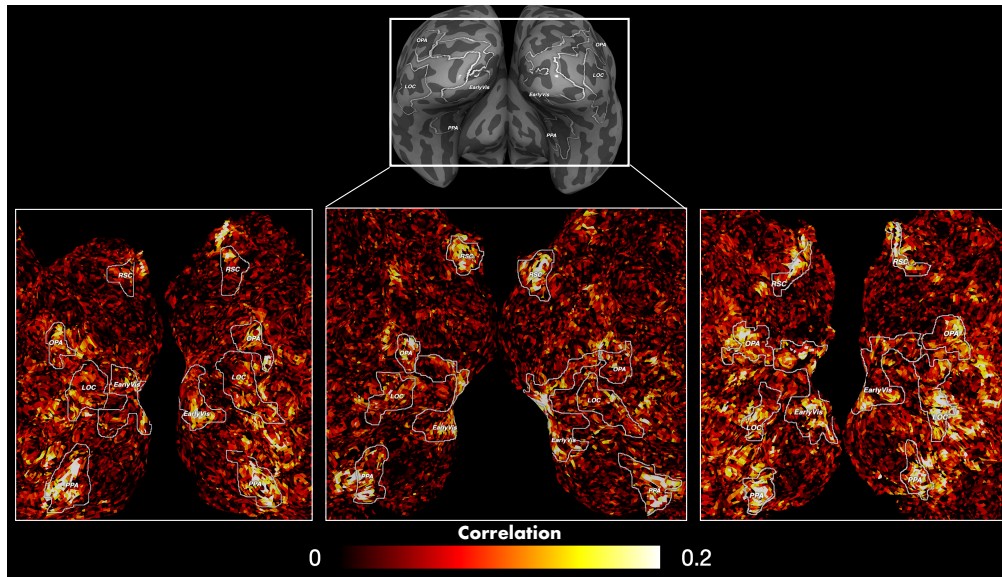

Figure 2: **Whole brain prediction correlation using task representation of scene classification network.** The flat maps are cropped from the occipital regions of the brain. The upper zoom-out view shows the relative locations of the flat maps. Lower colored figures are the prediction performance across 3 participants. Prediction results are consistent across subjects.

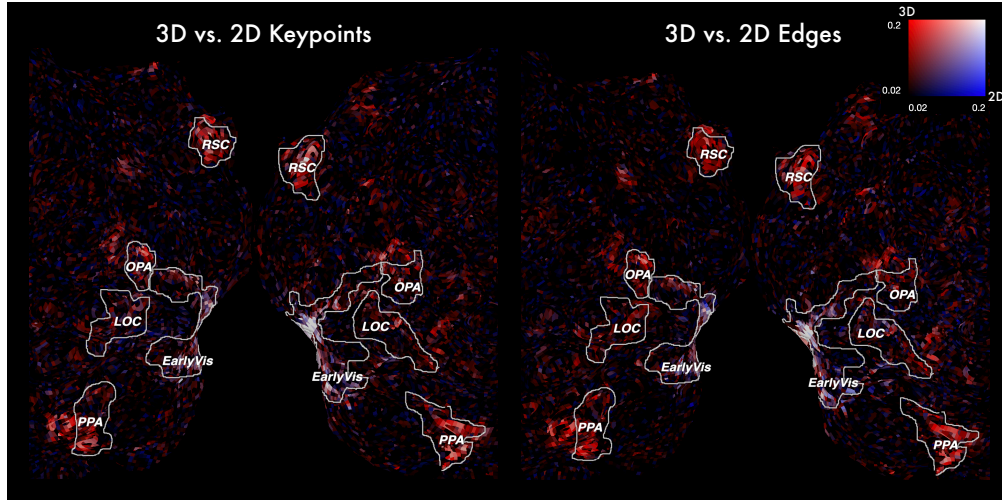

Figure 3: **Contrast of prediction performance (measured with Pearson correlation coefficients) between 2D and 3D features in one sample subject (subject 1).** The flat maps are cropped similar to as in Figure 2. The color map indicates the difference in correlation coefficients: red: 3D > 2D; blue: 2D > 3D. 3D task features predict better in scene selective regions and in more anterior parts of the brain.

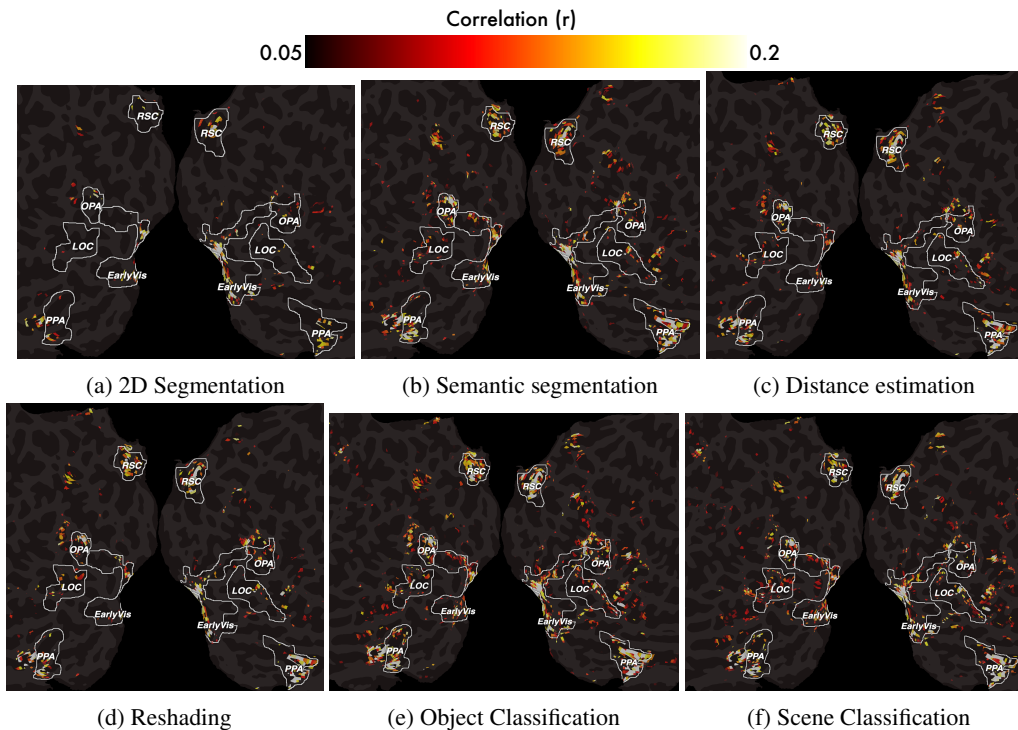

Figure 4: **Predictive voxels using tasks features from Taskonomy[20] in one sample subject (subject 1).** Predictive regions of different tasks differ from each other across tasks.

## 3.3  Evaluation of Neural Representation Similarity

To this point we have shown that the neural prediction maps across tasks differ from one another; at the same time, there are many overlapping voxels across the predicted regions. Importantly, this pattern of voxels as predicted by the tasks can be exploited and used to infer task relationships in the brain. We computed task similarity averaged across 3 subjects using the methods discussed

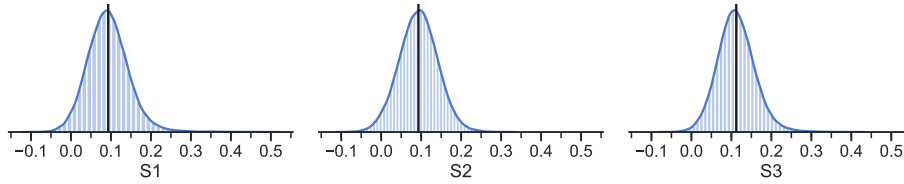

Figure 5: **Noise ceiling derived from cross subjection prediction.** Each subfigure is a histogram of correlation scores across voxels. The black lines on each subfigure indicate the average correlation values.

in 2.4 (Figure 6). The individual patterns of task similarity are almost identical across 3 subjects. More specifically, correlations (Pearson's r) between similarity matrices for each pair of subjects are: 0.9610 (S1 vs. S2), 0.9477 (S1 vs. S3) and 0.9407 (S2 vs. S3). In this comparison across the whole brain, tasks such as 2.5D segmentation, room layout estimation, surface normal estimation, scene classification etc. have similar predictions patterns.

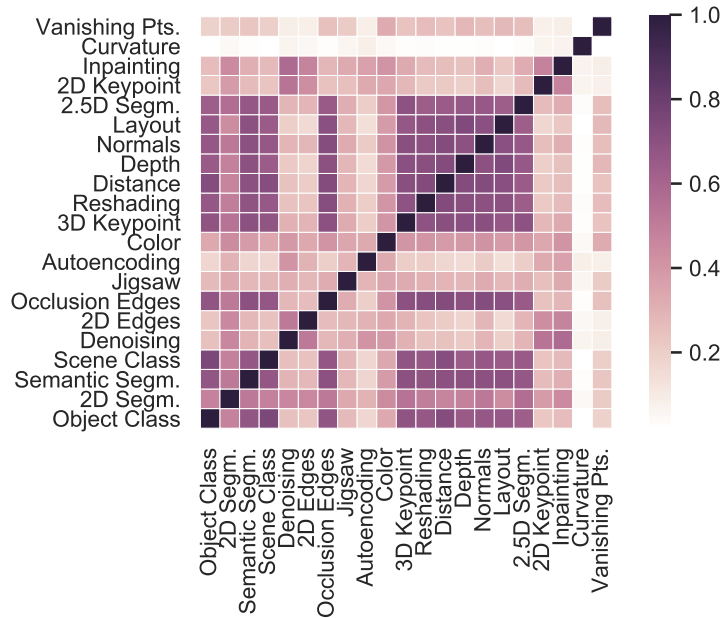

Figure 6: **Prediction similarity matrix across 21 tasks, averaged across 3 subjects.** A large similarity value between task X and Y indicates encoding models with features representation from task X and Y have similar predictions of brain responses.

## 3.4 Task Similarity Tree

To further explore the relationship between tasks as represented in the brain, we ran hierarchical clustering on the prediction correlation results and visualized the clustering results as dendrograms. Figure 7 compares the task similarity tree based on transferring-out patterns in the original Taskonomy paper[20], with the task similarity tree generated based on similarity in voxel prediction performance. Trees independently generated for each subject show great similarity. In the Taskonomy result, tasks are clustered into 3D (indicated in green), 2D (blue), low-dimensional geometric (red) and semantic (purple) tasks. Interestingly, the tree derived from brain representation also shows a similar structure: semantic, 2D and 3D tasks are clustered together. The differences between the two similarity trees

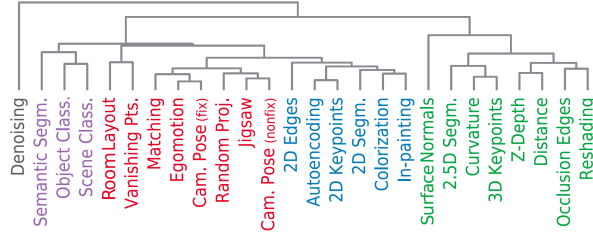

(a) Taskonomy from transfer learning

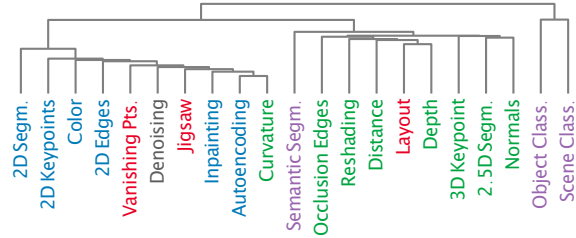

(b) Neural Taskonomy (subject 1)

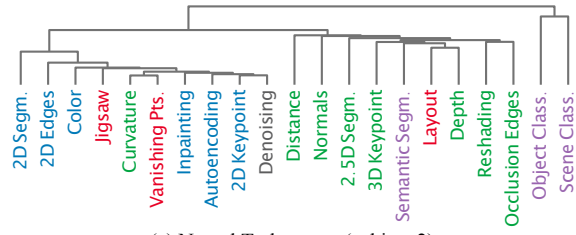

(c) Neural Taskonomy (subject 2)

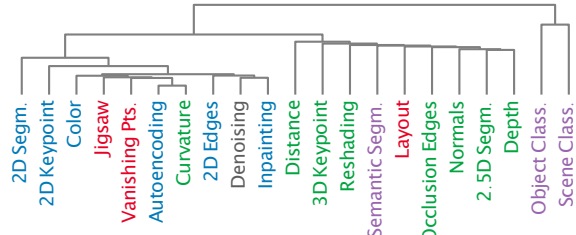

(d) Neural Taskonomy (subject 3)

Figure 7: **Task Trees from (a) Taskonomy[20] and (b-d) brain representation of tasks from 3 differnet subjects.** Tasks in (b-d) are colored according to colors in (a). Similar clusters of 2D (in blue), 3D (green) and semantic (purple) tasks are found among neural taskonomy trees. Clustering results are highly consistent across three subjects.

may be due to low absolute performance of the encoding model. For example, the model with features from curvature estimation task has less than 10 significant voxels in some subjects which may lead to bias in the representation of the task tree. Overall the similarity between two task trees shows that, at a coarse level, neural representation of task information is similar to that found through transfer learning. The clustering and dendrogram structures are stable across subjects and across different linkage criteria. Aside from using "average" linkage for clustering, as shown here, we also used "ward" linkage criterion (shown in the appendix) and obtained similar structures.

## 4   Discussion

The architecture of the primate visual system reflects a series of computational mechanisms that enable high performance for accomplishing evolutionarily adaptive tasks[27]. However, the precise

nature of these tasks remains unknown because of the limitations of neuroscience data collection methods and the lack of interpretability of intermediate visual representations. To address these issues we leveraged the space of vision tasks learned through transfer learning in Taskonomy[20] and the recent availability of a larger-scale human functional imaging dataset, BOLD5000[22]. One challenge we faced was the substantial difference between the image distributions of BOLD5000 (which contains general objects and scenes) and the Taskonomy dataset (which includes indoor scenes exclusively). As such, when we applied the pre-trained Taskonomy models to BOLD5000 images, we found that these models didn't perform as well as on the Taskonomy dataset, especially for the outdoor images used in BOLD5000. Such inconsistency in image distribution is unavoidably reflected in the encoding model performance and hinders us from making more specific claims about task spaces in the brain. One solution to this issue would be to use a more general computational model of visual tasks, as well as a larger brain dataset based on more images, both of which are outside of the scope of this paper.

In the future we would also like to investigate the unique and shared variance explained by each task. At present we are still unclear as to what transferability between tasks within Taskonomy predicts for similarity in task representations within the brain.

Finally, although our whole brain prediction maps do seem to suggest the involvement of additional functional brain areas beyond the pre-defined ROIs, we strongly feel that making claims about new functionally-defined brain areas would be premature given our current data and analysis. We believe that to make robust claims about new "functional territories", we would first need to run additional validation experiments in which specific manipulations are used to establish that specific brain regions are sensitive to the tasks in question.

## 5 Conclusion

Our results reveal that task-specific representations in neural networks are useful in predicting brain responses and localizing task-related information in the brain. One of the main findings is that features from 3D tasks, compared to those from 2D tasks, predict a distinct part of visual cortex. In the future we will incorporate features from other tasks to obtain a more comprehensive picture of task representation in the brain.

For years neuroscientists have focused on recovering which parts of the brain represent a given type of information. However, what are the computational principles behind the encoding of information in the brain? We observe feedforward hierarchies in the visual pathways, but what are the stages of information processing? To date, we have few satisfying answers. The ultimate goal in studying task representation in the brain is to answer some of these questions. We exploited the task relationship found in transfer learning and used it as a ground truth of visual information space to study the neural representation of visual and semantic information. In sum, our paper provides an initial attempt in using task relationships to answer broader questions of neural information processing.

## Acknowledgments

We thank the BOLD5000 team for providing the public accessible data. We would also like to thank Jayanth Koushik for the help with model fitting, paper edits and result visualizations, as well as Nadine Chang for useful discussions regarding the project.

## Footnotes

[1]All code is available on https://github.com/ariaaay/NeuralTaskonomy

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
