[Supplementary Material]

# Appendix

## 1 Cross subject prediction

Figure 1: Noise ceiling derived from cross subject prediction. Each subfigure is a histogram of correlation scores across all voxels in a subjects. Y-axis indicates which subjects the prediction is from while x-axis indicates which the prediction is of.

## 2 Model predictions of tasks from Taskonomy

## 3 Task similarity tree using "ward" linkage

Figure 2: Model predictions of tasks from Taskonomy. Voxels below significance threshold ($p \geq 0.05$, FDR corrected) are masked.

(a) Neural Taskonomy (Subject 1)

(b) Neural Taskonomy (Subject 2)

(c) Neural Taskonomy (Subject 3)