[Reviews · NeurIPS 2019]

Reviewer 1



Overall, the paper rests on a very clever idea "Instead of choosing a generic object-classification CNN as a source of visual features, we built encoding models with individual feature spaces obtained from different task-specific networks.", that deserves publication. It is indeed probably the most clever idea that I have seen on deep features encoding in brain imaging for years. The paper has relatively nice figures. I identified one technical issue: "To determine the significance of the predictions, we calculated the p-values of each correlation 109 coefficient and report false discovery rate (FDR) corrected p-values." -> these p-values are likely problematic, because correlations and structure in the data violate the hypothesis necessary to interpret correlations as p-values (iid etc.). 172-173 "Both Figure 2 and 3 show that prediction results are consistent across three participants or tasks despite anatomical differences in their brain structures." not sure that this statement is useful (it is hard to judge) or whether it is actually true. It is bad that 6 of the 25 tasks were nor used (or are not presented) in the paper: even if they do not seem meaningful, they may carry some information. If they do not, it is also an interesting point. the results are slightly disappointing in the sense that the process * yields overall low accuracy, * does not seem to uncover new functional territories * as a result of low accuracy, only gives functional tendencies ---not sharp conclusions. Fig.6 I think that data-derived tree structures are too brittle, and thus doe not support sharp conclusions. Fig.7 obviously the two graphs are not identical. There seems to be large modules in the fMRI graph not present in taskonomy graph. Interesting point on covariate shift across images in the discussion. typos "To better understand underlying the nature of the visual features" " et. cetra"

Reviewer 2



Having read the authors’ response and the other reviews, I think I may have gotten a little overly excited about this paper, but I still think it is innovative and significant. In fact, I found the relationship between the two clusters in Figures 6a and 6b more convincing than I did originally. However, I am dropping my score a notch because, while I really like this paper, I believe it is in the top 50%, not the top 15% of NIPS papers. This is again, because the analysis is not as rigorous as one would like. Also, after having read the author's response and the other reviews, I have edited my review a bit below. This paper is highly original, combining two very different efforts in a novel way. The BOLD5000 dataset has only recently come out, and is already a gargantuan effort, providing fMRI responses of multiple subjects to 5000 images of various types. This paper is the first I’m aware of that takes advantage of this enormous (for fMRI) dataset. And the use is quite clever: In Taskonomy, networks with similar architectures are trained to perform 26 different tasks, and then task similarity is based on how well the internal representations developed for each task transfer to other tasks. These same encodings are used here (from 19 tasks considered to be ones humans might perform in the course of processing images) with ridge regression to predict voxel activations. In other words, the 5,000 images used in BOLD5000 are given to the Taskonomy networks and the resulting encodings are used to predict the 5,000 BOLD activations of 55,000 voxels. This gives rise to a set of correlations between voxels and tasks. The results are in part interesting because across three subjects, for five different ROIs, the pattern of correlations between tasks and voxels are quite similar. It would have been useful, I think, if the authors had correlated the correlations instead of simply using ocular regression to compare them. Less exciting are the results in Section 3.2, where they apply the same technique to whole-brain voxels. It is difficult to interpret these results (in Figure 2). It is unclear what we are supposed to take away from this Figure. Also, it would probably be more visible if the background for these figures was white instead of black (ditto Figs 3 and 4 – Figure 1 from the appendix is better). I note here that the figures look much better in the pdf than in my printout of the paper. I couldn’t tell from the printout that Figure 2 had labeled the regions. Figure 3 was more informative, showing that 3D features give better correlations than 2D features, which is an interesting finding. I couldn’t see any blue voxels in my printout of Figure 3. Figure 4 would have been enhanced by some discussion of any insights to be gleaned from it. Figure 4 is more like, see what we can do? Some interpretation would have been good. The results in section 3.4 and Figures 5 and 6 are interesting. I would have left out of the paper section 3.5 in favor of more analysis/discussion of the results in Figure 5 and 6. Figure 7 is not at all convincing, and would be best left in, say, the supplementary material. The last two sentences of section 3.5 seem more like wishful thinking. It would have been interesting if there was a significant correlation between the graphs, using a randomization test. The paper is well-written and fairly clear. As the paper says, this paper “provides a first attempt in using predictive-map-generated task relationships to answer broader questions of neural information processing. 
” I agree, and I think this will lead to many fruitful future efforts along these lines. Very minor comments: line 66: it's either etc., or et cetera, not et. cetra. et is not an abbreviation for anything. It is what it is. line 119: networks-> network. line 167 and the caption of Figure 2. Line 167 suggests that Figure 2 shows the correlations for *each* feature space, but it only shows the correlations for scene classification. One issue about the task tree: to the extent that tasks are similar (as measured by the Taskonomy paper), won't their predictions be similar? This would seem to influence the brain taskonomy and overly inflate the similarity between the trees. lines 201 and 202 seem like a real stretch. Line 280: frmi in the bibtex -> {FMRI}

Reviewer 3



## Quality It seems a few pieces of crucial logic are missing from this study. First, it seems necessary to know the ceiling brain encoding performance in order to understand how adequate these models are as pictures of neural representation. How well could an optimal model of visual processing perform in brain encoding with this neuroimaging data? The authors could estimate a performance ceiling by attempting to learn an encoding model predicting one subject’s brain activations from another subject’s activations in response to the same stimulus, for example. Second, I’m also not clear on the logic of the latter analyses and comparison with Taskonomy. The authors claim (line 193) that, because the qualitative analyses on the brain encoding error patterns match those of the Taskonomy transfer patterns, “neural representation of task information is similar to that found through transfer learning.” I don’t think this inference is valid — the only valid inference would be about the model representations, not the neural representations. Specifically, if model A predicts brain activations similarly to model B, then it is likely to transfer across vision tasks similar to the way that model B transfers. The inference about neural representations is only licensed if we know that all of the variance in the neural activation space is explained by model activations, I think. We can’t know this without an estimate of ceiling performance for this dataset. ## Clarity Some (minor) methodological and presentational things are not clear to me after reading the paper: 1. How is cross-validation of the regression regularization hyperparameter fit across subjects? You should ideally use nested cross validation here to reduce the chance of overfitting. 2. BOLD5000 provides images at multiple time offsets after stimulus presentation — are you using the standard average of TR3–4 or something different? Why? 3. Figure 4: There is no interpretation of this figure in the text. (PPA seems to differ in correlation; otherwise I can’t see anything qualitatively important by squinting at these maps.) Also, I assume this is derived from a single subject — which? Do the qualitative patterns (whichever you want to pick out) replicate across subjects? 4. Figure 5: why is the diagonal not always equal to 1? (Are there multiple training runs of the same model input to this analysis? Or maybe this is averaging across subjects?) 5. I don’t understand what the utility of figure 7 is — it seems to be conveying the same message as figure 5, but in a less readable way. ## Significance Modulo some of the things I see as logical issues with the paper stated above, I think the claimed result (contribution #3 in my review) would be quite interesting if it were substantiated by the data. ## Originality This is an interesting combination of existing methods.

[Author Response · NeurIPS 2019]

We thank the reviewers for their useful feedback.

**R1&R3, Accuracy baseline and ceiling:** P-values under the assumption of independence can be problematic, but
here: (1) We have fMRI events that are separated by 10 seconds. (2) Before training, we z-scored within sessions and
runs. We trained and predicted with shuffled, unordered single trials, but not a time series of data. (3) The BH FDR
controls the false discovery rate under positive dependence which is a very common assumption for fMRI. To further
minimize assumptions on the underlying distributions, we ran permutation test where we shuffled responses 5000 times,
computed the correlation scores, and obtained FDR corrected $p$-values for both ROI results (shown in figure below for
Subject 1; * = FDR corrected $p < 0.00001$) and whole brain results.

Across the whole brain, $p$-values obtained through permutation tests show more significant voxels across tasks. To
provide a better estimate of the variance ceiling, we ran ridge regression to predict between subjects. In the figure below
we show the prediction correlation for each subject from other two subjects (each subfigure is a histogram of correlation
scores across voxels). The average correlation between predictions and true responses across voxels for each subject
are: 0.0931, 0.0932, 0.112, as shown by the black lines on each plot (this includes low SNR voxels that are not engaged
   by the task). The accuracy we obtained on the significant voxels across the task is close to the ceiling.

**R1, Other Taskonomy tasks:** Out of the 25 tasks that are provided by the Taskonomy pre-trained task bank, 4 of them
take multiple images as input and therefore are excluded in this analysis since brain responses are to single images. For
the 21 single image tasks, 2 of them (jigsaw and inpainting) were not included in the paper since they are less human
related. Nevertheless, the features from these tasks still encode important information and significantly predict both
whole brain voxels and ROIs, as shown in first figure above. Because of limited space, we do not include the whole
brain prediction map.

**R1-3**, To quantify the consistency of results across subjects, we computed correlations of prediction accuracy across
subjects: 0.7957 (S1 vs. S2), 0.9034 (S1 vs. S3) and 0.9345 (S2 vs. S3). It would be more difficult to quantify
consistency outside of ROIs since structurally projecting one whole brain onto another blurs functional data. We also
computed correlations of task distance matrices across subjects: 0.9128 (S1 vs. S2), 0.9304 (S1 vs. S3) and 0.8884 (S2
vs S3). Tree structures of tasks can be obtained through hierarchical clustering on these pairwise distance matrices.
Therefore the high correlations across subjects reinforce the **stability of task tree structures (R1)**. **(R2&R3)** Task
similarity does not necessarily imply similar predictions in the brain. It depends on the degree of overlap between tasks
- if they have very high overlap they will predict basically the same brain data. But if tasks overlap less (say 50% of the
variance), then there is plenty of "room" for each task to predict unique brain areas based on the non-shared variance.

**R3&R1** For all encoding models, we predict an average of TR3 and TR4. Regression regularization hyperparameters
are fit for individual voxels and individual subjects. To avoid overfitting, only validation data but not test data is used to
pick the hyperparameters. The task similarity matrix presented in Figure 5 in our paper is an average of distances from
3 subjects (using cosine similarity). Curvature and autoencoding have < 1 similarity to themselves because for some
subjects they predict zero significant voxels (zero vectors is undefined for cosine distance, and 0 was returned). We also
used other similarity metrics and the patterns were very similar. **(R1)** Prediction maps in the whole brain do predict
novel areas beyond the pre-defined ROIs. However, making claims about new functionally-defined brain areas would
be premature given our current data and analyses - to the detriment of the field, such claims are often made based on
inconclusive data (so-called "fishing expeditions"). To make a robust claim about new "functional territories" we expect
to first run additional validation experiments in which specific, theory-driven manipulations were used to establish that
specific brain regions are sensitive to the task in question.

[Meta-Review · NeurIPS 2019]

This paper takes a creative idea of using the feature spaces from many different computer vision task networks to help understand fMRI data from different brain regions. Reviewers find the idea interesting but the results are not particularly clear/definitive or surprising.